# Modification of Alumina Inclusions in SWRS82B Steel by Adding Rare Earth Cerium

**Yi Wang** [1,2]**, Changrong Li** [1,2,]*****, Linzhu Wang** [1,2,]*****, Xingqiang Xiong** [1,2]**, Lu Chen** [1,2] **and Changling Zhuang** [1,2]

[1] School of Materials and Metallurgy, Guizhou University, Guiyang 550025, China; wy465717296@163.com (Y.W.); Xxq15223673790@163.com (X.X.); chen841283779@163.com (L.C.); clzhuang@gzu.edu.cn (C.Z.)

[2] Guizhou Key Laboratory of Metallurgical Engineering and Process Energy Conservation, Guiyang 550025, China

***** Correspondence: crli@gzu.edu.cn (C.L.); lzwang@gzu.edu.cn (L.W.)

**Abstract:** The aluminum oxide inclusions in SWRS82B steel seriously affect the drawing performance of the steel strand. In this study, the influence of different additions of cerium (within the range of 0–0.034%) on the composition, morphology, size, number, and distribution of alumina inclusions was studied by scanning electron microscope and energy spectrum analyzer. The evolution of the composition of inclusions with different cerium additions was calculated based on classical thermodynamics and Factsage software calculation. The thermodynamic calculated results were consistent with the experimental results. It indicates that the modification route of $Al_2O_3$ inclusions in SWRS82B steel by increased cerium additions is as follows: $Al_2O_3 \rightarrow Ce_2S_3 + CeAlO_3 + Ce_2O_2S + Al_2O_3 \rightarrow Ce_2S_3 + CeAlO_3 + Ce_2O_2S/Ce_2S_3 + Ce_2O_2S \rightarrow Ce_2S_3 + Ce_2O_2S$. Besides, when the amount of cerium is in the range of 0.023% to 0.030%, $CeAlO_3$ inclusions gradually disappear. The best characteristics of inclusions in this study were obtained in experimental samples with cerium addition of 0.023%, in which the minimum size of inclusions is in the range of 3.52–4.84 μm and mostly uniform distribution. Finally, the mechanism on the modification by cerium was discussed based on the composition evolution of inclusion during solidification with Factsage calculation and experimental results. The compositions of inclusions were also analyzed based on the inclusion evolution model.

**Keywords:** alumina inclusions; inclusion modification; rare earth elements; cerium oxide (sulfur) compounds

## 1. Introduction

With the actual needs of automobile high-speed rail and construction steel, higher requirements are put forward for the strength and toughness of metal materials. High drawing strength steel has become the focus of current research. The wire rod will break during the drawing process, affecting the performance of the product. The drawing limit is mainly related to the carbon content and structure of the steel [1]. SWRS82B is a high-strength steel that is widely used in wire ropes and steel strands and it is mainly used in the construction and transportation industries. It has developed rapidly in recent years [2]. A large amount of research shows that [3–5] it is important to pay attention to the size, morphology, and deformation of non-metallic inclusions for improving the performance and service life of SWRS82B high-strength steel wire rope and steel strand steel.

Aluminum has such a strong deoxidizing ability that it is often used as a deoxidizer in the steelmaking process. However, the addition amount of aluminum has a great influence on the quality of the molten steel, which is likely to cause nozzle clogging. Calcium treatment is used to modify

alumina inclusion, decreasing the detrimental effect of alumina [6]. Many scholars have proposed that calcium treatment can transform high-melting alumina inclusions into low-melting liquid composite inclusions. Although calcium treatment can alleviate the problem of nozzle clogging, liquid inclusions have no capability for providing heterogeneous nuclei that can provide structure control [7], and the notch effect depends on the shape factor and volume fraction. Ye et al. [8] proposed a layer-by-layer reaction model for $Al_2O_3$ modification by calcium in the order of $Al_2O_3$—$CA_6$—$CA_2$—$CA$—$CA_X$(liq), and, as the activity of $Al_2O_3$ decreases, the CaS from the outermost chromatograph prevented the modification reaction from proceeding further, resulting in the alumina inclusions not being completely modified into a liquid state. Rare Earth Metals (REM) compares very favorably with calcium regarding ease of dissolution [9]. REM has a very high affinity for both oxygen and sulfur. REM can also form oxysulfides $(REM)_2O_2S$, which are more stable than any of the other sulfides [10]. Therefore, it could be expected that REM is capable of reaching almost all oxygen and sulfur, whether in solution or in some form of less stable inclusions [11]. Magnesium treatment can reduce the number and size of the inclusions in steel, and the volatilization of magnesium can play a stirring role. However, the formation of magnesium aluminum spinel inclusions in steel is harmful to the quality of the steel. Therefore, it is necessary to look for a new solution. It is found that cerium has a more obvious effect on the purification of molten steel. Spherical inclusions with smaller sizes formed in the steel with cerium addition and they can be used as a nucleating agent in the molten steel, refining grain. Hirata [12] and Yang [13] et al. found that cerium can modify the alumina inclusions with higher hardness in cerium oxysulfide with lower hardness. Gao et al. [14] found that the spherical inclusions generated and the inclusion size decreased in IF steel with 0.02% cerium due to the transformation from alumina inclusions into Ce-Al-O-S inclusion. They also proposed that $Ce_2O_3$ and $Ce_2O_2S$ with hexagonal crystal systems have a low degree of mismatch with δ-Fe. Wen et al. [15] found that the type of inclusions in SS400 low-carbon steel can be controlled by changing the S/O ratio and the cerium content. Li et al. [16] found that when the amount of added cerium into low-carbon high-manganese steel is up to 0.034%, the size of the inclusions tends to increase and the modification effect decreases. Wang et al. [17] found that when the amount of added cerium to spring steel exceeds 0.017%, all the inclusions in the molten steel will be completely modified into rare earth inclusions. Li et al. [18] found that spherical and liquid inclusions were generated in rare earth-treated 253MA steel at 1500 °C. The size of the inclusions has a greater influence on the fatigue resistance of the material, and the inclusions with small size and low hardness do little harm to the fatigue fracture of the steel, which is beneficial to the service life of high-carbon steel.

There are many thermodynamic studies on the treatment of alumina inclusions with rare earth in molten steel [19–27]. However, limited works are focused on the rare earth treatment of high-carbon steel. The mechanism on the evolution of the inclusions modified by rare earth in high-carbon steel needs to be analyzed. The effect of the amount of added cerium on the compositions and characteristics of inclusions was studied in SWRS82B steel. The modification process and transformation mechanism on inclusions in SWRS82B steel treated by cerium were discussed based on classical thermodynamic calculations and Factsage software calculation. This study will provide a reference for solving the problem of modification of alumina inclusions in high-carbon steel.

## 2. Materials and Methods

### 2.1. Experimental Materials and Procedures

An intermediate frequency induction furnace (Wangxin Precision Industry CO. LTD., Guangzhou, Guangdong, China) was used for the experiment. The molten steel was prepared by melting industrial pure iron (purity 99.5%. % represents mass percentage, hereinafter), recarburizer (C ≥ 98.5%, S ≤ 0.05%), and Fe-68%Mn alloy in the intermediate frequency induction furnace (170 mm OD × 150 mm ID × 280 mm HT) for melting. The capacity of the alumina crucible used in this experiment was 20 kg. The total weight of the materials placed in the crucible in each experiment was 7 kg. When the

intermediate frequency induction furnace was heated to 1873 K, the materials were stirred to completely melt. After 10 min, the Al bar was added to deoxidize (Al ≥ 98%, Si ≤ 0.6%, Fe ≤ 0.7%) and stir the molten steel. After 10 min, cerium particles (Purity 99.9%) were added and the molten steel was stirred. Five minutes later, the molten steel was poured into the dry mold coated with talcum powder to cool and demold. Table 1 shows the chemical composition of the experimental SWRS82B steel. In the first furnace, only aluminum flakes were added during the smelting, and no cerium was added. The second furnace sample was smelted with 0.008% cerium particles. The third furnace sample was smelted with 0.023% cerium particles. The fourth furnace sample was smelted with 0.034% cerium particles. The cerium yield was very low. The final cerium contents in each sample and yield are shown in Table 2. The carbon, silicon, manganese, phosphorus, sulfur, and aluminum content were measured by inductively coupled plasma emission spectrometry (ICP), and the total oxygen content was measured by inert gas fusion pulse-infrared absorption spectroscopy.

**Table 1.** Chemical composition of the test steel (weight precent/%).

| Element | C | Si | Mn | P | S | Al | O |
|---------|-------|------|------|-------|-------|--------|--------|
| S1 | 0.826 | 0.21 | 0.83 | 0.019 | 0.008 | 0.0254 | 0.015 |
| S2 | 0.823 | 0.21 | 0.82 | 0.018 | 0.007 | 0.0253 | 0.0093 |
| S3 | 0.824 | 0.20 | 0.83 | 0.019 | 0.007 | 0.0254 | 0.0053 |
| S4 | 0.821 | 0.21 | 0.83 | 0.019 | 0.009 | 0.0254 | 0.0068 |

**Table 2.** Actual cerium contents and measured yields.

| Sample | Actual Content/%Ce | Actual Yield/% |
|--------|--------------------|----------------|
| S1 | 0 | 0 |
| S2 | 0.008 | 9.34 |
| S3 | 0.023 | 10.07 |
| S4 | 0.034 | 9.53 |

*2.2. Sample Processing*

Samples (10 mm × 10 mm × 10 mm) were taken from the center of cylindrical steel ingot and ground with SiC sandpaper from 400 mesh to 2000 mesh. The surface of the samples was cleaned with ethanol. A German Zeiss ΣIGMA+X-Max20 (Baden-Wurttemberg, Germany) scanning electron microscope (SEM) with energy dispersive spectrometer (EDS) was used to analyze the size, morphology, and chemical composition distribution of the inclusions. Two hundred eighty-nine consecutive SEM pictures were taken under 1000 times magnification, corresponding to a total area of 4.6 mm × 4.6 mm. Image-ProPlus image processing software (Image-Pro Plus6.0, Rockville, Media Cybernetics, MD, USA) was used to analyze the size, number, and distribution of the inclusions on the surface of the photographed sample.

## 3. Results and Discussion

*3.1. Change in Composition of Inclusions*

SEM and EDS were used to observe the inclusions on the sample and the results are shown in Figures 1 and 2. The typical inclusions of sample S1 and their map scanning are shown in Figure 1. The type of inclusions detected by SEM/EDS is alumina inclusions. It can be seen from Figure 1a–g that there are many alumina inclusions with various morphology in sample S1. The distribution of elements in typical inclusions in samples S2–S4 are shown in Figure 2. The element distribution of typical inclusion in sample S2 is shown in Figure 2a,b. When the amount of cerium is 0.008%, the contents of Ce and S in the outer layer of typical inclusions are higher and the content of Al in the inner layer is relatively higher. The inclusion with cerium oxysulfide wrapping $Al_2O_3$ formed, indicating that the $Al_2O_3$ inclusions are modified by cerium. The morphology of the inclusions is still irregular in S2 with

0.008% Ce and it suggests that the amount of added cerium is insufficient. The element distribution of typical inclusion in sample S3 is shown in Figure 2c,d. There are mainly cerium oxysulfides with less $Al_2O_3$ in sample S2 with 0.023% Ce. The element distribution of typical inclusion in sample S4 is shown in Figure 2e,f. When the addition amount of cerium is 0.034%, no composite inclusions with $Al_2O_3$ as the core were detected. It was observed that the size of the inclusions increased slightly compared to the sample S3. It can be concluded that the inclusions are mainly large-particle irregularly shaped $Al_2O_3$ inclusions in the steel with no cerium addition. To further confirm the composition of Ce-Al-O-S inclusions, the main constituent elements were homogenized. The inclusions are mainly $Ce_2S_3 + Ce_2O_2S + CeAlO_3 + Al_2O_3$ composite inclusions in the steel with cerium addition of 0.008%. The main inclusions are composed of $Ce_2S_3 + Ce_2O_2S + CeAlO_3$ and $Ce_2O_2S$ in the steel with a cerium addition of 0.023%. When the amount of cerium is 0.034%, the main inclusions are $Ce_2S_3 + Ce_2O_2S$ type inclusions in the steel with cerium addition of 0.034%. The addition of cerium has a modification effect on the alumina inclusions. With the increase of the cerium content, the transition route of inclusions is as follows: $Al_2O_3 \rightarrow Ce_2S_3 + CeAlO_3 + Ce_2O_2S + Al_2O_3 \rightarrow Ce_2S_3 + CeAlO_3 + Ce_2O_2S/Ce_2S_3 + Ce_2O_2S \rightarrow Ce_2S_3 + Ce_2O_2S$.

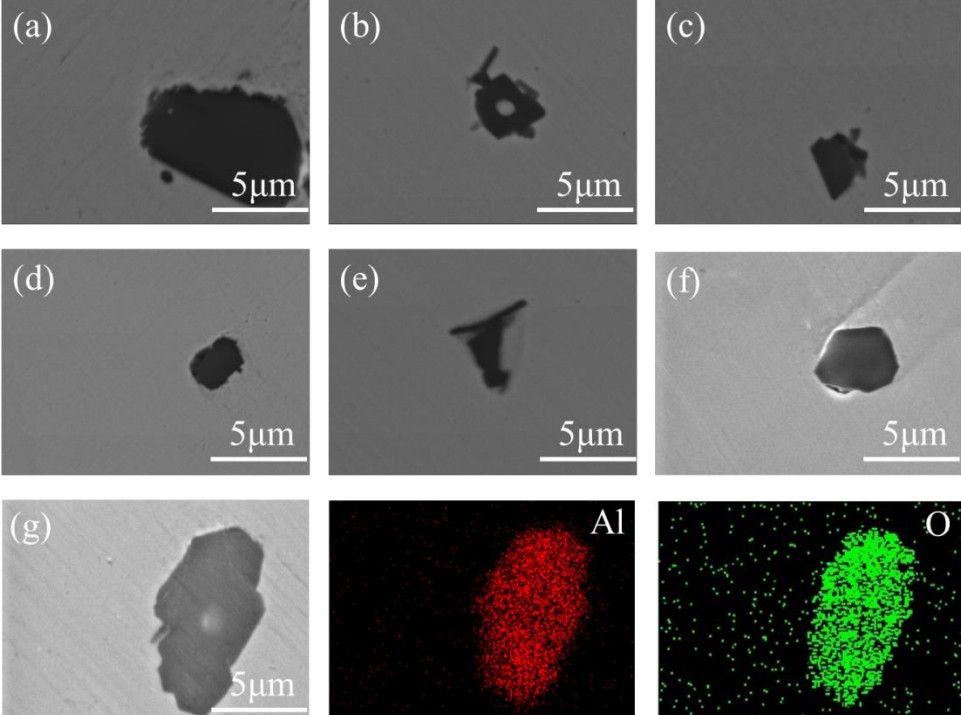

**Figure 1.** SEM/EDS image of alumina inclusions in the sample S1. (**a**–**g**) SEM image of alumina inclusions in the sample S1.

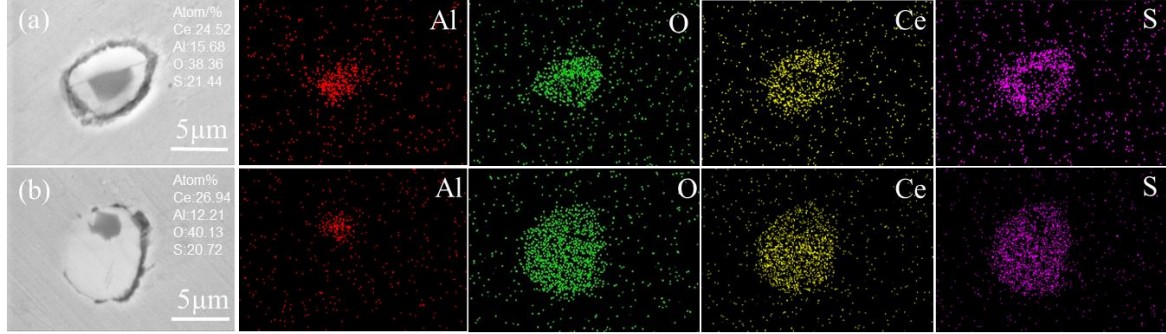

**Figure 2.** *Cont.*

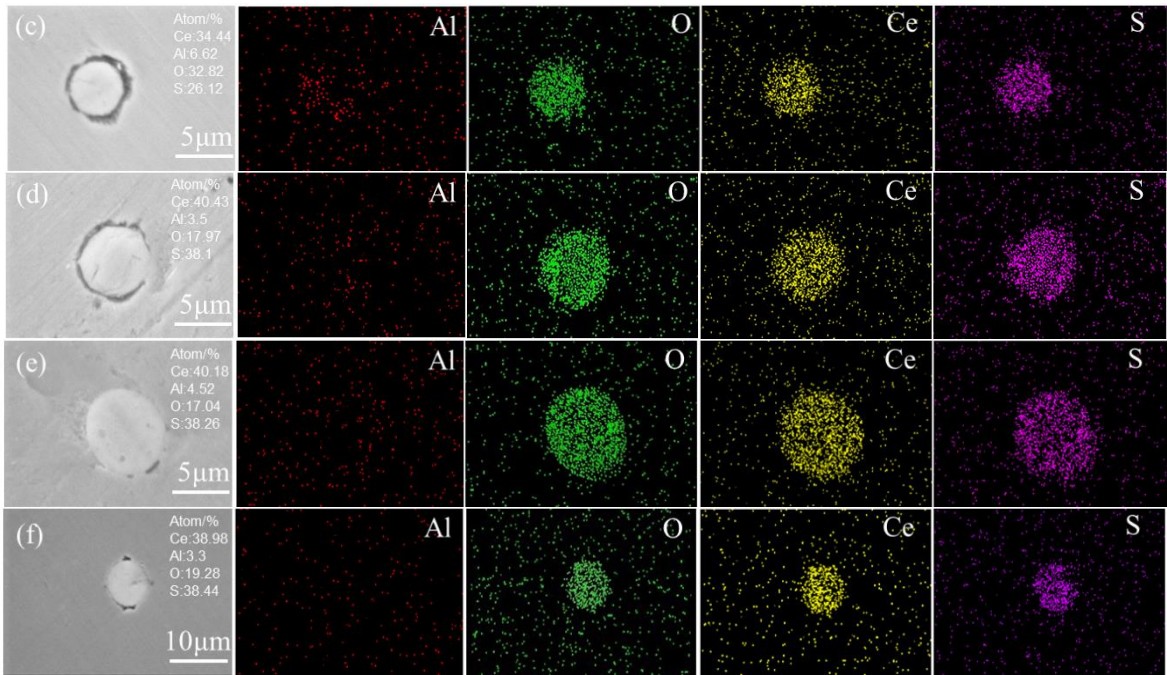

**Figure 2.** SEM/EDS image of typical inclusions in the sample S1–S4. (**a**,**b**) sample S2, (**c**,**d**) sample S3, (**e**,**f**) sample S4.

### 3.2. Number, Size, and Area Density of Inclusions

The size and number density distribution of the inclusions was obtained using a scanning electron microscope and Image-ProPlus image processing software as shown in Figure 3. In Figure 3a, the number of inclusions smaller than 2 μm is gradually increasing with the increase of cerium, indicating the inclusions are refined. The number of inclusions larger than 10 μm gradually decreases, and the inclusions with a size larger than 10 μm were only detected in samples S1 and S2 samples. The number density of inclusions in all these four samples was in the range of 166–258 cm$^2$ as shown in Figure 3b. The number of inclusions in sample S2 is the largest with a number range of 209–258 cm$^2$, while the number of inclusions in sample S1 is the smallest with a number range of 166–192 cm$^2$. In Figure 3c, the average size range of inclusions is reduced from 8.65–11.32 μm to 3.52–4.84 μm, and then increased to 6.01–7.5 μm, and the average size of the inclusions in sample S3 is the smallest.

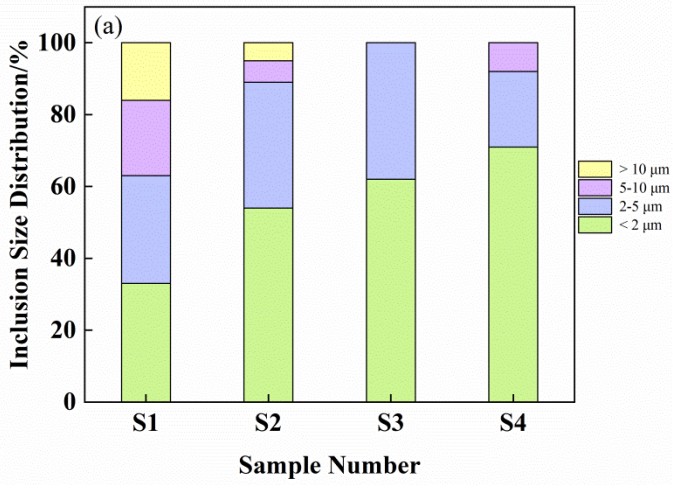

**Figure 3.** *Cont.*

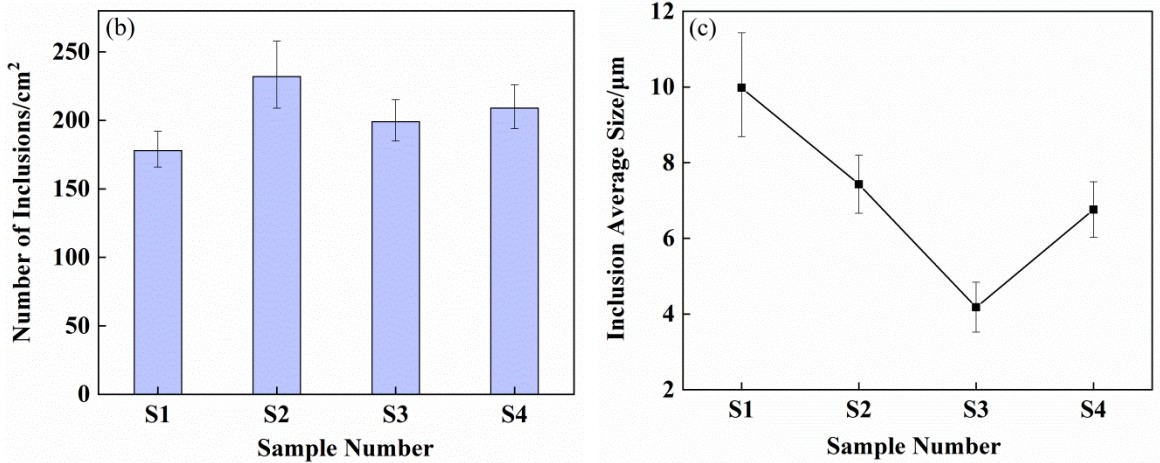

**Figure 3.** The number and average size of inclusions. (**a**) inclusion size distribution, (**b**) number of inclusions, (**c**) inclusion average size.

In order to observe the distribution trend of the inclusions in the steel more intuitively, the area density distribution of the inclusions is analyzed by Image-ProPlus software, as shown in Figure 4. In Figure 4a, the distribution of the inclusions is uneven. There are inclusions with large size and the maximum areal density in Sample S1 accounts for 1.1%. It can be seen from Figure 4 that the distribution of inclusion in samples tend to be more uniform, as follows: S1 < S2 < S4 < S3. The area density of the inclusions in sample S1 is large and the distribution is uneven, mainly because the inclusions in sample S1 are mainly alumina inclusions. The alumina inclusions are not easily wettable with molten steel, resulting in the aggregation of inclusions to form large-size inclusions. In addition, the force between the alumina inclusions is large, and the inclusions are easily attracted to each other, resulting in an uneven distribution of inclusions. With the increase of the cerium content in the inclusions, the growth rate of the inclusions slows down. Because alumina is gradually modified into cerium oxysulfide, the attractive force between inclusions becomes weak and their distribution is more even. The main factor affecting the area density and distribution of the inclusions is the interactive force acting on the inclusions, which is related to the composition and size of inclusions.

Based on the data in Figure 3b,c, the "measured volume of inclusions" (average volume x number) of samples S1–S4 are as follows: 56,626 $\mu m^3$–145,753 $\mu m^3$, 32,165 $\mu m^3$–74,446 $\mu m^3$, 4177 $\mu m^3$–9975 $\mu m^3$, 22,040 $\mu m^3$–49,897 $\mu m^3$. Through comparison, there seems to be a good agreement between Figures 3 and 4 and oxygen contents in Table 1. Of course, these values only show the approximate inclusion content. The relationship between total oxygen content and inclusion volume needs to be explored more deeply, but they should be comparable anyway.

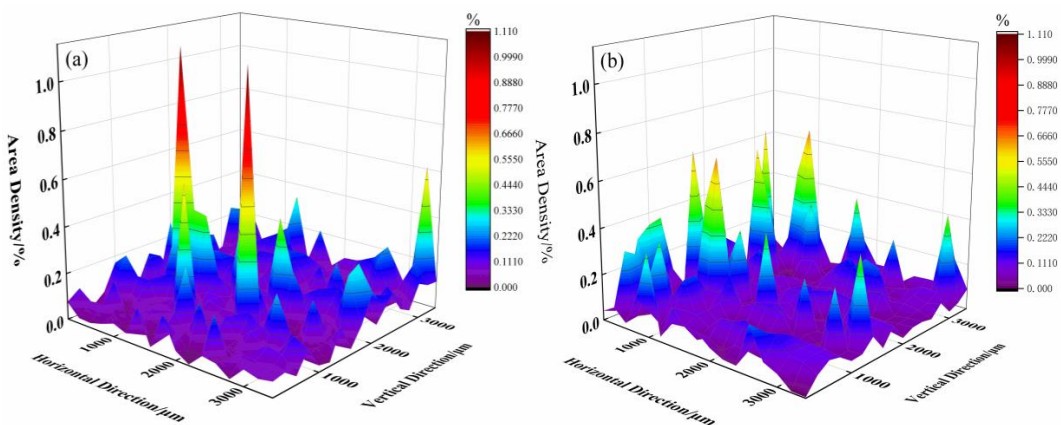

**Figure 4.** *Cont.*

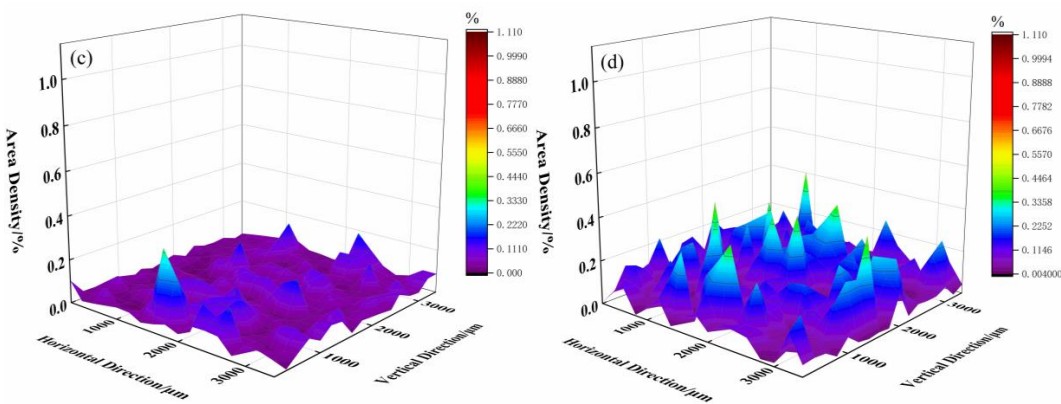

**Figure 4.** Density distribution of the inclusions on the cross-section. (**a**) Sample S1 area density distribution, (**b**) Sample S2 area density distribution, (**c**) Sample S3 area density distribution, (**d**) Sample S4 area density distribution.

### 3.3. Thermodynamic Calculation of Inclusions

To determine the evolution mechanism of the inclusion with alumina modified by cerium, we should consider the formation of inclusions in the actual reaction process. The interaction coefficients of elements in the SWRS82B steel at 1873 K are shown in Table 3 [28]. According to the Wagner's model formula in Equation (1), the calculated activity coefficient of [O], [S], [Ce], and [Al] element, and then according to the activity formula in Equation (2), the calculated activities of [O], [S], [Ce], and [Al] are shown in Table 4. The reactions of inclusion transformation and the standard Gibbs free energy of these reactions in molten SWRS82B steel are listed in Table 5 [29–33].

$$\lg f_i = \sum_{j}^{n} e_i^j w(j\%) \tag{1}$$

$$a_i = f_i \cdot w(i\%) \tag{2}$$

where $f_i$ activity coefficient, $w(i\%)$ and $w(j\%)$ represent the mass percentages of $i$ and $j$, $e_i^j$ is the activity interaction coefficient of $i$ on $j$, and $a_i$ represents the activity of substance $i$ degree.

**Table 3.** Interaction coefficients of elements in the SWRS82B steel at 1873 K [28].

| $e_i^j$ | C | Si | Mn | P | Al | O | S | Ce |
|---|---|---|---|---|---|---|---|---|
| O | −0.45 | −0.133 | −0.021 | 0.07 | −3.9 | −0.20 | −0.133 | −0.57 |
| S | 0.11 | 0.063 | −0.026 | 0.029 | 0.035 | −0.27 | −0.026 | - |
| Ce | −0.077 | - | 0.13 | 1.77 | −2.58 | −106 | −10.32 | 0.0069 |
| Al | 0.091 | 0.0056 | - | - | 0.045 | −6.6 | 0.03 | - |

Note: $i$ = O, S, Ce, Al; $j$ = C, Si, Mn, P, Al, O, S, Ce.

**Table 4.** Activities of [O], [S], [Ce], and [Al] in all samples used at 1873 K.

| Number | $a_{[O]}$ | $a_{[S]}$ | $a_{[Ce]}$ | $a_{[Al]}$ |
|---|---|---|---|---|
| S1 | 0.00454 | 0.00967 | 0 | 0.02419 |
| S2 | 0.00452 | 0.00964 | 0.00718 | 0.02625 |
| S3 | 0.00450 | 0.00965 | 0.02163 | 0.02801 |
| S4 | 0.00438 | 0.00965 | 0.03142 | 0.02737 |

**Table 5.** Reaction equation and the standard Gibbs free energy [29–33].

| Number | Equation | $\Delta G^{\theta}/(\text{J·mol}^{-1})$ |
|--------|----------|----------------------------------------|
| 1 | $2[Al] + 3[O] = Al_2O_3(s)$ | $-1,225,196 + 393.78T$ |
| 2 | $[Ce] + 2[O] = CeO_2(s)$ | $-852,720 + 249.96T$ |
| 3 | $[Ce] + 3/2[O] = 1/2Ce_2O_3(s)$ | $-714,380 + 179.74T$ |
| 4 | $[Ce]+[O] + 1/2[S] = 1/2\ Ce_2O_2S(s)$ | $-675,700 + 165.5T$ |
| 5 | $[Ce] + 3/2[S] = 1/2Ce_2S_3(s)$ | $-536,420 + 163.86T$ |
| 6 | $[Ce] + 4/3[S] = 1/3Ce_3S_4(s)$ | $-497,670 + 146.3T$ |
| 7 | $[Ce] + [S] = CeS(s)$ | $-422,100 + 120.38T$ |
| 8 | $[Ce] + 3[O] + [Al] = CeAlO_3(s)$ | $-1,366,460 + 364.3T$ |

When the temperature is 1873 K, according to the classical thermodynamic calculation method, the formation of inclusions in the actual reaction process is calculated. Combining Tables 4 and 5 and the Equations (3) and (4), we calculated the actual active product and equilibrium active product of inclusions in samples S1–S4, and judged the actual active product in steel. If the ratio between actual active product and balanced active product is greater than 1, then the inclusions meet the forming conditions, and the calculation results are shown in Figure 5 (values with ratios greater than 1 in Figure 5 are written as 1). Without considering MnS, the only possible substances in sample S1 are $Al_2O_3$. The possible substances in sample S2 are $Al_2O_3$, $CeAlO_3$, and $Ce_2O_2S$. The possible substances in sample S3 are $Al_2O_3$, $CeAlO_3$, and $Ce_2O_2S$. The possible substances in sample S4 are $Al_2O_3$, $Ce_2O_2S$.

$$\Delta G^{\theta} = -RT\ln k \tag{3}$$

$$\varepsilon = a_A/a_B \tag{4}$$

where $\Delta G^{\theta}$ is the Gibbs free energy of the reaction system under standard conditions, R is the molar gas constant, the value is 8.314, T is the temperature, $k$ is the reaction equilibrium constant, $\varepsilon$ is the activity ratio, and $a_A$ is the activity of the actual reaction process, $a_B$ represents the activity in the equilibrium state.

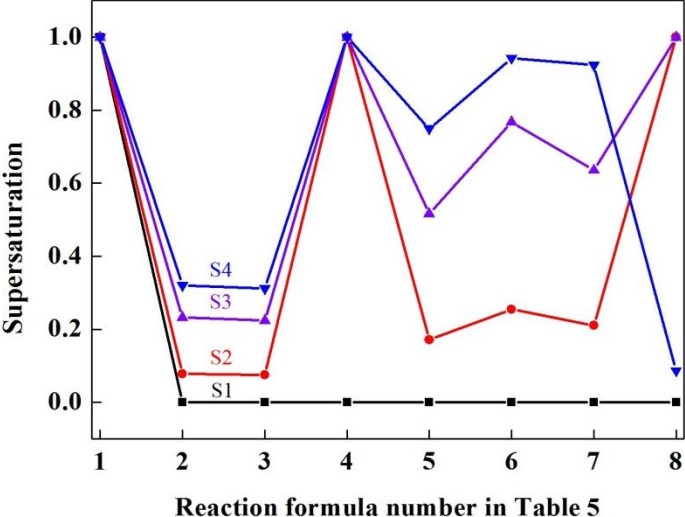

**Figure 5.** Comparison chart of activity product and equilibrium activity product of different reaction formulas at 1873 K.

According to the composition of samples S2–S4 in Table 1, Factsage software (FACTSAGE7.2, Thermfact/CRCT and GTT-Technologies, Montréal and herzogenrath, Canada and Germany) was used to calculate the equilibrium. When the temperature is 1873 K, the evolution of the inclusion composition with different amounts of cerium is shown in Figure 6. In Figure 6a–c, as the content of cerium increases,

three types of inclusions $Ce_2O_3 \cdot 11Al_2O_3$, $CeAlO_3$, and $Ce_2O_3$ are generated. In Figure 6a, the weight percentages of oxygen, sulfur, and aluminum used in the calculation are 0.0093%, 0.007%, and 0.0253%, respectively. $Ce_2O_3 \cdot 11Al_2O_3$ and $CeAlO_3$ formed when the cerium content is 0.008%, $Ce_2O_3 \cdot 11Al_2O_3$ inclusions may decompose into $Al_2O_3$ and $Ce_2O_3$ inclusions. In Figure 6b, the weight percentages of oxygen, sulfur, and aluminum used in the calculation are 0.0053%, 0.007%, and 0.0254%, respectively. $CeAlO_3$ is formed when the cerium content is 0.023%. In Figure 6c, the weight percentages of oxygen, sulfur, and aluminum used in the calculation are 0.0068%, 0.009%, and 0.0254%, respectively. $Ce_2O_3$ formed when the cerium content is 0.034%. However, no $CeAlO_3$ inclusions were found with SEM, which may be because the content of $CeAlO_3$ inclusions is lower than $Ce_2O_3$ inclusions, $CeAlO_3$ can continue to combine with oxygen and sulfur to form $Ce_2O_2S$, and $CeAlO_3$ inclusions are difficult to find. Therefore, combined with the actual amount of cerium during the experiment, the inclusions are transformed according to the following route: $Al_2O_3 \rightarrow Al_2O_3 + CeAlO_3 + Ce_2O_3 \rightarrow CeAlO_3 + Ce_2O_3 \rightarrow Ce_2O_3$. Since there is no data for cerium oxysulfide in the Factsage software database, the transition route of inclusions needs to be modified in conjunction with classical thermodynamic calculations. Combining the classical thermodynamic results and Factsage thermodynamic results, the composition of inclusions in SWRS82B steel at 1873 K transforms with cerium addition as follows: $Al_2O_3 \rightarrow Al_2O_3 + CeAlO_3 + Ce_2O_2S \rightarrow CeAlO_3 + Ce_2O_2S \rightarrow Ce_2O_2S$.

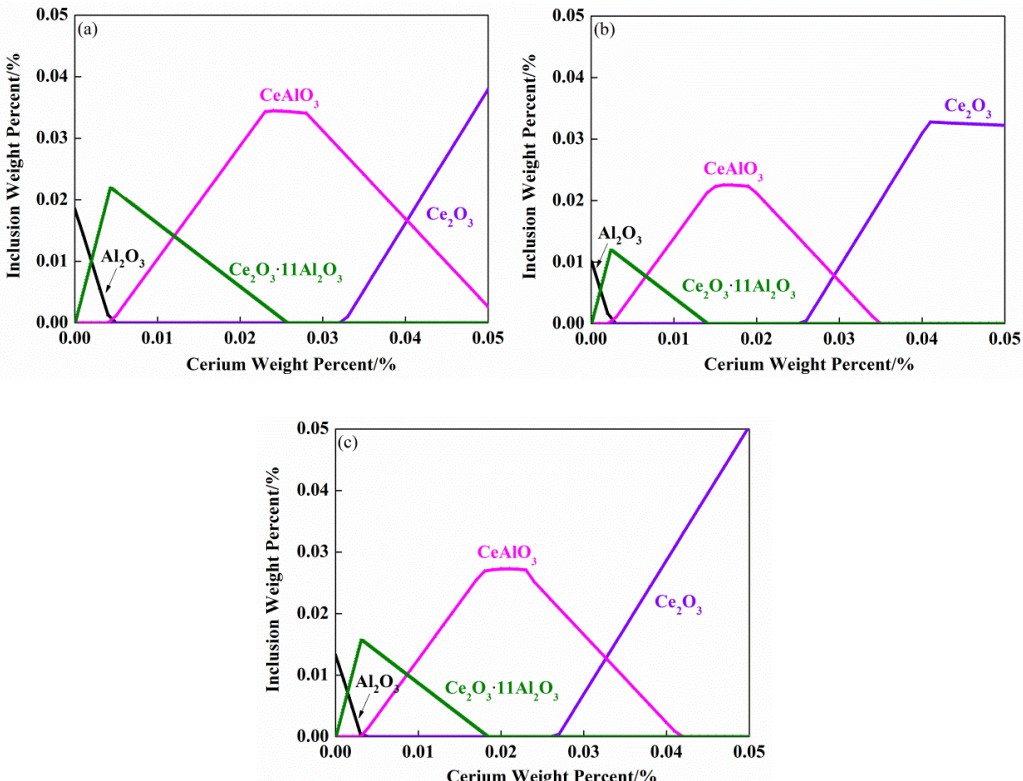

**Figure 6.** Evolution process of $Al_2O_3$ inclusions when adding cerium at 1873 K. Calculations performed for (**a**) sample S2 steel composition; (**b**) sample S3 steel composition; (**c**) sample S4 steel composition.

### 3.4. Transformation of Inclusions during Cooling and Solidification

According to the composition of molten steel, as shown in Tables 1 and 2, Factsage software (FACTSAGE7.2, Thermfact/CRCT and GTT-Technologies, Montréal and herzogenrath, Canada and Germany) is used to calculate the equilibrium state of different cerium additions and the evolution of inclusions composition at different temperatures, as shown in Figure 7. In Figure 7a, the weight percentages of oxygen, sulfur, and aluminum used in the calculation are 0.015%, 0.008%, and 0.0254%. In Figure 7b, the weight percentages of oxygen, sulfur, and aluminum used in the calculation are

0.0093%, 0.007%, and 0.0253%. In Figure 7c, the weight percentages of oxygen, sulfur, and aluminum used in the calculation are 0.0053%, 0.007%, and 0.0254%. In Figure 7d, the weight percentages of oxygen, sulfur, and aluminum used in the calculation are 0.0068%, 0.009%, and 0.0254%. At 1873 K, when cerium is not added to the SWRS82B steel, the inclusions were mainly composed of $Al_2O_3$. MnS inclusions began to precipitate at 1633 K(1360 °C) during the solidification of molten steel. The reaction is described by reactions (5) and (6), as shown in Figure 7a. At 1873 K, when the sample steel is added with 0.008% cerium, the inclusions are mainly $CeAlO_3$ inclusions. During the solidification of molten steel, when the temperature is 1643 K (1370 °C), the $CeAlO_3$ will partly decompose into $Al_2O_3$, while the original content of $CeAlO_3$ will decrease, and MnS inclusions begin to precipitate. When $CeAlO_3$ disappears at about 1170 °C, $Ce_2S_3$ inclusions begin to precipitate. During solidification, the inclusions change according to the following route: $CeAlO_3 \rightarrow CeAlO_3 + Al_2O_3 + MnS \rightarrow Al_2O_3 + MnS + Ce_2S_3$, the reaction is described by reactions (5)–(7), as shown in Figure 7b. At 1873 K, when the sample steel is added with 0.023% cerium, the inclusions are mainly composed of $Ce_2O_3$ and $CeAlO_3$. When the temperature is 1713 K (1440 °C), the concentration of $Ce_2O_3$ will decompose into $Ce_2S_3$ and $CeAlO_3$, while the original content of $Ce_2O_3$ decreases, and the content of $Ce_2S_3$ and $CeAlO_3$ increases, $CeAlO_3$ disappears completely when it decreases to 1383K (1110 °C). During the solidification process, the inclusions are transformed according to the following route: $CeAlO_3 \rightarrow CeAlO_3 + Ce_2S_3 \rightarrow Ce_2S_3 + Al_2O_3$, the reaction is described by reactions (5)–(8), as shown in Figure 7c. At 1873 K, when the sample steel is added with 0.034% cerium, the inclusions are mainly composed of $Ce_2O_3$. As the temperature decreases at about 1673 K (1400 °C), $Ce_2S_3$ and $CeAlO_3$ inclusions start to precipitate successively. During the solidification process, the inclusions are transformed according to the following route: $Ce_2O_3 \rightarrow CeAlO_3 + Ce_2S_3 + Al_2O_3$, the reaction is described by formula (5)–(9), as shown in Figure 7d.

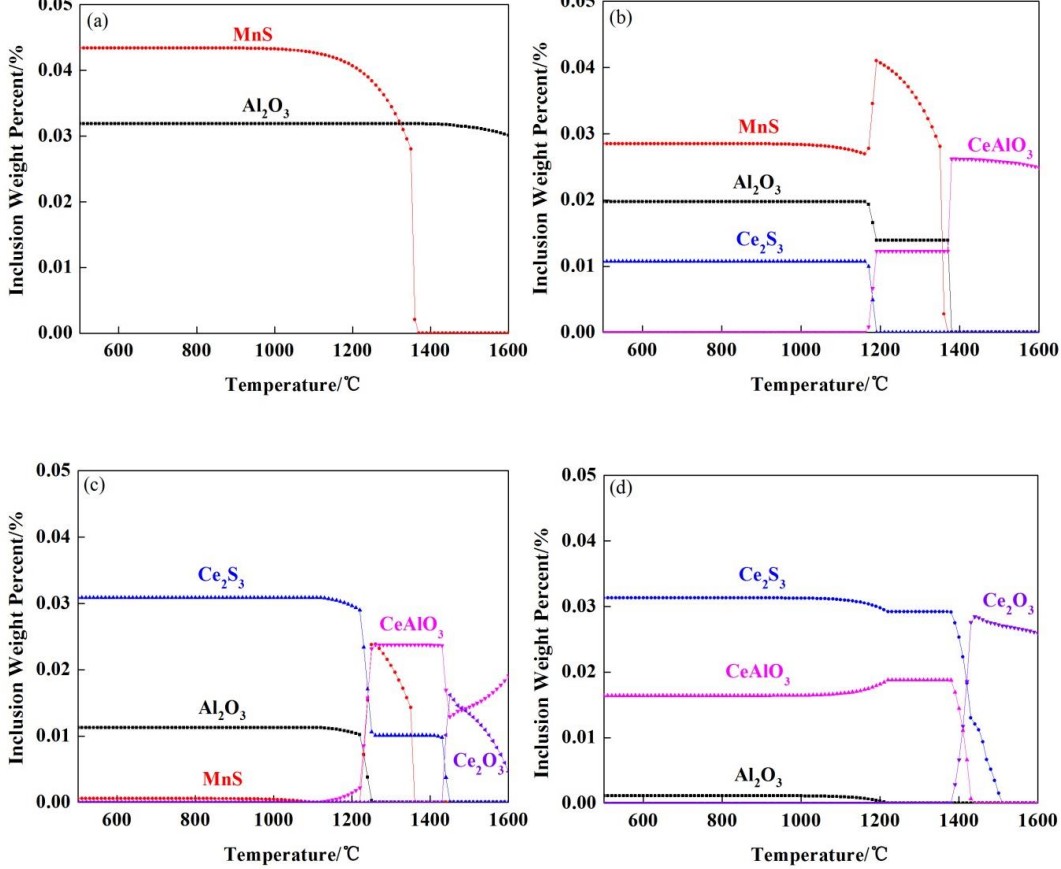

**Figure 7.** Calculation results by thermodynamic software. Effect of cerium addition on inclusions, (**a**) w(Ce%) = 0; (**b**) w(Ce%) = 0.008%; (**c**) w(Ce%) = 0.023%; (**d**) w(Ce%) = 0.034%.

$$2[Al] + 3[O] = Al_2O_3(s) \tag{5}$$

$$[Mn] + [S] = MnS \tag{6}$$

$$2[Ce] + 3[S] = Ce_2S_3(s) \tag{7}$$

$$Ce_2O_3(s) = 2[Ce] + 3[O] \tag{8}$$

$$[Ce] + Al_2O_3 = CeAlO_3 + [Al] \tag{9}$$

*3.5. Inclusion Evolution Model*

Combining the SEM/EDS results and the relevant calculation results in Figures 5–7 for analysis, the evolution of inclusions can be roughly divided into three routes according to different cerium additions, as shown in Figure 8. (I) Add a small amount of cerium to the molten steel, and the cerium reacts with $Al_2O_3$ to form $CeAlO_3$ inclusions. Due to the incomplete reaction, it also contains $Al_2O_3$ inclusions. As the reaction proceeds, $Ce_2O_2S$ inclusions formed in the outer layer. The reaction is shown in reactions (10)–(11). During the solidification process, $Ce_2S_3$ inclusions precipitated in the outer layer of inclusions. According to reactions (1)–(3), the thermodynamically stable order of inclusions that may be formed in liquid steel is $Ce_2O_2S > Ce_2S_3$. It can be seen that $Ce_2S_3$ is not stable, and may combine with sulfur and oxygen to form $Ce_2O_2S$. The reaction is described by reaction (12). Compared with the alumina inclusions before modification, the morphology and size of the modified inclusions have changed significantly. The average size of the composite inclusions is reduced in the range of 2 μm–3.12 μm, the morphology changes to nearly spherical. (II) When 0.023% cerium is added to the molten steel, the core $Al_2O_3$ of the composite inclusion nucleation is completely covered by $CeAlO_3$ inclusions. This may be due to the fact that the reaction in reaction (10) is relatively completed and converted to other substances, and $Ce_2S_3$ inclusions formed during the solidification process. Compared with the morphology of alumina inclusions, the edges and corners of the composite inclusions gradually degenerate into a smooth spherical surface. The average size of the composite inclusions is reduced in the range of 5.13 μm–6.48 μm compared with the $Al_2O_3$ inclusions. The fine dispersion effect is the best. (III) When 0.0034% cerium is added to molten steel, $Ce_2O_2S$ inclusions formed at 1873 K, and $CeAlO_3$ and $Ce_2S_3$ inclusions formed during the solidification process. The reaction is described by reactions (10)–(13). Because the reaction did not reach a completely ideal equilibrium state, the actual results and the calculated results are within the expected deviation, which is basically consistent with the experimental results.

$$[Ce] + Al_2O_3 = CeAlO_3 + [Al] \tag{10}$$

$$6[Ce] + 3[S] + 2Al_2O_3(s) = 3Ce_2O_2S(s) + 4[Al] \tag{11}$$

$$Ce_2S_3 + 2[O] \rightleftharpoons Ce_2O_2S + 2[S] \tag{12}$$

$$Ce_2O_3 + [S] \rightleftharpoons Ce_2O_2S + [O] \tag{13}$$

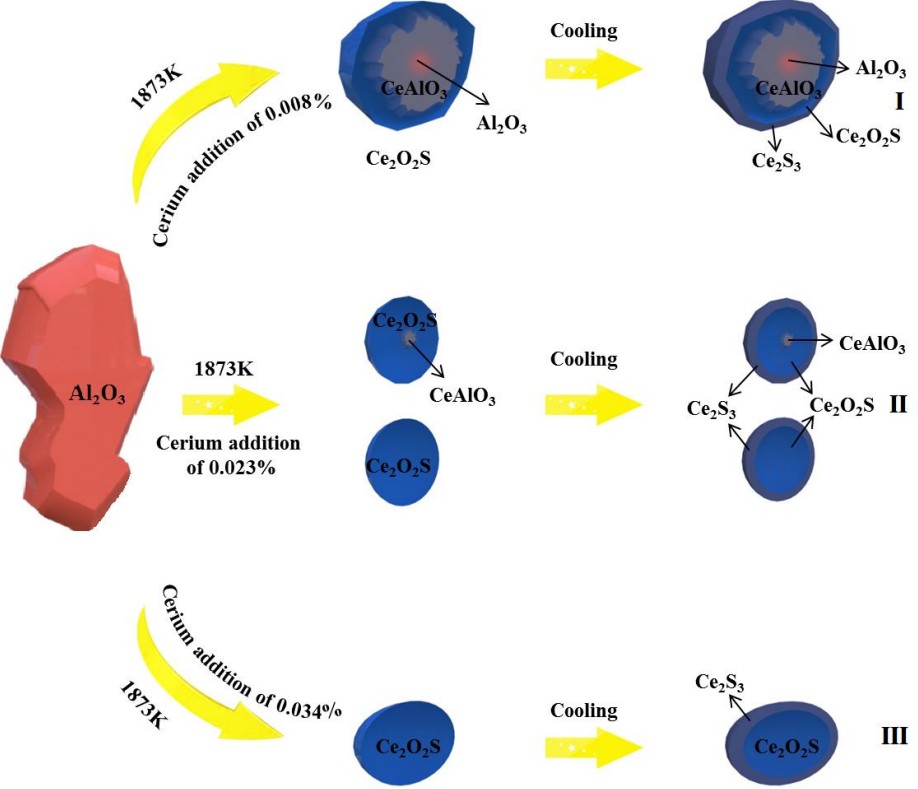

**Figure 8.** Inclusion evolution model.

## 4. Conclusions

The influence of the addition of cerium in SWRS82B in the range of 0–0.034% on the modification of alumina inclusions was studied. Through experiments and theoretical calculations, the conclusions are as follows:

(1) Before the addition of cerium, the average size of the inclusions is in the range of 8.65 μm–11.32 μm and the shape is irregular. When cerium is added, the inclusions gradually become spheroidized and the average size of the inclusions varies in the range of 3.52 μm–8.19 μm, and if the cerium content exceeds 0.023%, the inclusions will grow excessively.

(2) Compared with $Al_2O_3$ inclusions in sample S1, the average size of inclusions produced by adding 0.023% cerium is reduced in the range of 5.13 μm–6.48 μm, and the best characteristics of inclusions in this study were obtained in experimental samples with cerium addition of 0.023%, in which the minimum size of inclusions is in the range of 3.52 μm–4.84 μm and most uniform distribution.

(3) The classical thermodynamic calculation results are basically consistent with the experimental composition results, and the transition route of inclusions in SWRS82B steel at 1873 K is as follows: $Al_2O_3 \rightarrow Ce_2O_2S + CeAlO_3 + Al_2O_3 \rightarrow CeAlO_3 + Ce_2O_2S/Ce_2O_2S \rightarrow Ce_2O_2S$.

(4) $Ce_2S_3$ precipitated during solidification. The modification route of $Al_2O_3$ inclusions in SWRS82B steel by increased cerium additions is as follows: $Al_2O_3 \rightarrow Ce_2S_3 + CeAlO_3 + Ce_2O_2S + Al_2O_3 \rightarrow Ce_2S_3 + CeAlO_3 + Ce_2O_2S/Ce_2S_3 + Ce_2O_2S \rightarrow Ce_2S_3 + Ce_2O_2S$.

**Author Contributions:** Conceptualization, Y.W. and C.L.; methodology, Y.W., L.W.; software, Y.W., X.X., and L.C.; writing—original draft preparation, Y.W. and L.W.; writing—review and editing, L.W. and C.L.; funding acquisition, C.L. and C.Z. All authors have read and agreed to the published version of the manuscript.

**Funding:** This project is s financially supported by the National Science Foundation of China with the grant No. 51864013, 52074095, 51704083, and 52064011.

**Conflicts of Interest:** The authors declare no conflict of interest.

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
