# Peer review of "Modification of Alumina Inclusions in SWRS82B Steel by Adding Rare Earth Cerium"

_metals, doi:10.3390/met10121696_

Round 1

Reviewer 1 Report

Thanks a lot for considering the comments from side of the reviewers in every detail. The quality of the paper is now much better and the approaches have become very clear. Congratulations.

Author Response

Thank you for your recognition of the article, and I wish you all the best and success in your work.

Reviewer 2 Report

The article concerns the current topic related to modification of alumina inclusions in steel by adding cerium. The introduction was very well prepared, taking into account the review of the research to date and the justification for taking up the topic. The indicated research methodology does not raise any objections. The test results are presented graphically and in tables. Certain comments require correction:
- Figure 1 - a-f - no description
- line 154-156 - not clear
Fig 3a - legend not readable
- Fig. 4 - description difficult to read
- line 184 and 188 - literature first [35] then [29-34], it should be the other way around
- Fig. 5 - caption under the figure in Table 3, should be in table 5
- equation [8] is not well balanced
- Fig. 8 - description requires improvement, especially with Ce (currently not readable)

Author Response

Dear reviewer,

On behalf of my co-authors, we thank you very much for giving us an opportunity to revise our manuscript. We appreciate editor and referees very much for their constructive comments and suggestions on our manuscript.

Our revised manuscript (Manuscript number: 999366) entitled “Modification of alumina inclusions in SWRS82B steel by adding rare earth cerium” is shown in attachment.

We have studied the valuable comments from you and reviewers carefully and have made revision which marked in red and light blue in the paper. The point to point responds to the reviewer’s comments are listed in “Detailed description of the changes in the revised manuscript”.

Thank you very much for your attention and consideration. We truly hope that our work can be accepted in your publication.

Sincerely yours,

Yi Wang

Reviewer 3 Report

Dear Authors,

your paper Modification of alumina inclusions in SWRS82B steel 2 by adding rare earth cerium discusses the effect of Ce on Al2O3-based non-metallic inclusions modification.

The paper is interesting because gives an alternative process to the more traditional Ca-treatment for the modification of Al2O3-based inclusions and reports indicative amount of Ce to be added to obtain the best inclusions transformation

The paper is well organized and couples experimental and simulation results to explain the mechanismes induced by Ce addition

Being the topic hot, I suggest the publication of your research on Metals journal.

However, in the current state, there are some parts needing changes. In the pdf attached you can find some highlighted parts needing corrections/modifications or check.

In addition, a comparison in terms of inclusions chemistry and size distribution with a steel treated with Ca is warmly suggested. In this way, the benefit of RE treatment will be better highlighted.

Furthermore, there is not any mechanical characterization (even a traditional tensile test of Charpy impact test) to confirm that the 0.023% Ce addition lead to the best size, chemistry and morphology of the modified NMI. Again I warmly suggest to include a mechanical investigation and maybe a comparison with a tradition Ca-treated steel can be made also from this point of view.

Thus, I can reconsider the paper suitable for publication only after the addition of at least mechanical characterization

Best Regards

Author Response

(The authors gave the same response as above.)

Round 2

Reviewer 3 Report

Dear Authors,

thank you for having submitted a revised version of your paper

The most of the reviewers' comments were well addressed

However, I have still some minor changes to ask to yourself before a finale acceptance of your paper for publication in Metals

Please, find in attachment my comments directly on the pdf file

Besr Regards

Author Response

Dear editor,

On behalf of my co-authors, we thank you very much for giving us an opportunity to revise our manuscript. We appreciate editor and referees very much for their constructive comments and suggestions on our manuscript.

Our revised manuscript (Manuscript number: 999366) entitled “Modification of alumina inclusions in SWRS82B steel by adding rare earth cerium” is shown in attachment.

We have studied the valuable comments from you and reviewers carefully and have made revision which marked in green in the paper. The point to point responds to the reviewer’s comments are listed in “Detailed description of the changes in the revised manuscript”.

Thank you very much for your attention and consideration. We truly hope that our work can be accepted in your publication.

Sincerely yours,

Yi Wang

This manuscript is a resubmission of an earlier submission. The following is a list of the peer review reports and author responses from that submission.

Round 1

Reviewer 1 Report

Please correct English of your article.

1) Page 2, line 65. “Bin et al [15] studied …” Please correct this reference according to reference list: “Wen et al [15] studied …”.

2) Page 2, lines 89-90. “…alloy (pig iron), recarburizer (C≥98.5%, S≤0.05%), Fe-55%Si alloy, Fe-68%Mn alloy, aluminum bar (Al≥75%), rare earth particles (≥99.9 %) Put it into the alumina crucible …”

Please add composition of pig iron used in these experiments.

For what you used carburizer (C≥98.5%) in the melt, if pig iron contains usually >4% of C?

“…aluminum bar (Al≥75%)…” - Which other components (≤ 25%) are contained in aluminum bar?

“…rare earth particles (≥99.9 %) …” - ≥99.9 % of Ce or (Ce+La+…)?

According to the given description, rare earth particles was added into crucible and heated up to 1600oC together with other charge components. Is it correct or not? Please explain under which atmosphere (air, argon …) you carried out experimental melting.

3) Pages 2 and 3, lines 95-96. “…The inclusions are homogenized….” How and which inclusions (non-metallic inclusions?) were homogenized in the melt?

4) Page 3, Table 1. Please explain why O content decreased drastically in steel samples from experiment 1# to experiment 4#.

5) Page 3, lines 110-111. “…grind a certain section with SiC sandpaper from 400 mesh to 2000 mesh, wet grinding with diamond polishing agent,…” According to the given description, water was used during grinding and polishing of steel samples. Is it correct? Please describe this point in details because all Ce-contained inclusions can easy react with water. As a result, the obtained results can be wrong.

6) Page 6, lines 138-139. “1# sample is treated with rare earth cerium, and the size is relatively large and the morphology is irregular.” Is it correct? According to Table 2, the 1# sample does not Ce-treated.

7) Page 6, lines 144-145 and 149. “... The size is reduced compared to Figure 2.” and “... the size of the inclusions is reduced compared to Figure 3,…” Comparison of inclusion sizes only based on photographs shown in Figures 2, 3 and 4 is incorrect. It can be done based on results of measurements of large number of inclusions in each steel sample.

8) Page 6, line 145-146. “… When the addition amount of cerium is 0.023%, no Al2O3 inclusions are detected,…” It is not clear because Al2O3 was detected inside of inclusions, as shown in Figure 4. Please specify what you mean.

9) Page 6, line 153-154. “…and the inside All the inclusions wrapped in alumina disappeared.” It is incorrect because Figure 5a can be seen some Al content in the given inclusion at the higher brightness:

10) Page 7, Figure 6. How many inclusions were measured in each sample? According to data shown in Figure 6b, the number of measured inclusions n was too small (38-44 per sample). Therefore, the discussions and conclusions of this study are dubious and uncertain.

Value of n can be calculated as follows: n = NA ∙ Aobs, where NA is the number of inclusions per unit area given in Figure 6b and Aobs is the observed area of steel sample (0.2116 cm2).

Please add bars of standard deviation (σ) for average size of inclusions in Figure 6c.

11) Page 8, line 189-190. “…the force between the alumina inclusions is large, and the inclusions are easily attracted to each other,…” Please explain which force between inclusions you mean. How this force depends on the inclusion composition?

12) Page 8, lines 194-196. “…When the cerium content is too large, the growth rate of the inclusions increases with the increase of the cerium content concentration, the size of the inclusions becomes larger, and the surface density of the inclusions increases.” According to this sentence and data shown in Figure 6b and c, the number and size of inclusions in sample 4 increased compared to those in sample 3. However, contents of O and S in sample 4 are significantly smaller compared to those in sample 3. Please explain this disagreement.

13) Page 8, Table 3. Please add reference for each reaction in this table.

14) Page 9, Table 4. Please explain why you selected the given values for eOCe=-0.57 and eCeO=-5.03 and did not use other interaction coefficients. For instance, the Japan Society for Promotion of Science (JSPS) recommends for thermodynamic calculations the following values*: eOCe=-64, eCeO=-560, eSCe=-9.1and eCeS=-40. These values are very different compared with yours values and can significantly effect on results of thermodynamic calculations.

* - “Thermodynamic data for steelmaking”, edited by M.Hino and K.Ito, 19th Committee in Steelmaking, Tohoku University, 2010, Japan.

15) Page 9, Equations (2)-(4). What is J in equations (2) and (3)? What is K in equation (4)? Where is Q (the reaction entropy, line 217) in equations?

16) Page 10, lines 234-235. “…Combined with the test, further research found that when a[Ce] / a[Al] ≥ 0.08, Al2O3 inclusions can be modified into CeAlO3 inclusions” According to this sentence, Al2O3 inclusions can be modified into CeAlO3 inclusions with increasing of Ce content in steel samples (because the activity of Ce increases with an increased Ce content). Is it correct? Please check it. The activity of Ce is significantly larger than the activity of Al in your experiments.

17) Figure 10. Please change a color for Al2O3 in this figure because phases of Al2O3 (blue) and Ce2O2S (blue) look very similar in this figure.

18) Please specify a novelty of your article compared to previous publications** which focused on study of modification of Al-contained inclusions by addition of Ce or other rare earth metals (modification mechanism, changes of composition, size and number of modified inclusions, etc.).

** - For instance: Q. Wang, L.J. Wang, Y.Q. Liuc, K.C. Chou: "USING Ce TO MODIFY INCLUSION IN SPRING STEEL", J. Min. Metall. Sect. B-Metall., 53 (3) B (2017), 365-372. DOI:10.2298/JMMB161117026W

Reviewer 2 Report

The literature review focusses mainly on publications from China/Japan. In fact, the use of Ce for grain refinement in steel has a long tradition in Europe and US and the role of Ce with respect to the modification of alumina is also addressed. For the sake of completeness you should add the one or other of these publications.

2.1 Experimental: The master alloy should be pure iron and not pig iron. Some of the sentences between lines 88 and 117 are unclear. Please revise this part.

3. Results: The authors selected two representative inclusions from every sample for their discussion and conclusions.

Yes: Automated SEM/EDS is challenging for Ce-containing steels due to different gray scale differences. But statistical data on the change of the volume fraction or number density of certain types of inclusions between samples #1, #2, #3 and #4 is needed proove the evidence of the conclusions of the authors. Otherwise the reader has to believe that all inclusions in #3 are AlCeO3 or Ce2O2S and all inclusions in #4 are in the system Ce2O2S and no more Al-containing inclusions can be found in #4. This is unlikely. If the data is not available, the authors need to comment on the change of the inclusions population on a statistical basis.

In lines 135 - 159 some sentences are unclear, examples:

138 1# sample is treated with rare earth cerium

156 inclusions are mainly Ce2O2S-Al2O3 composite type The main inclusions

3.3 Thermodynamics

Fig. 9b indicates, that even for 0 % of Ce, AlCeO3 is the thermodynamically stable phase. In fact, Al2O3 should be the stable phase.

The use of FactSage and its databases gives a more complex view on the existing phases: For low-Ce it should be Al2O3, then with an increasing Ce-content it is Al11O18Ce + Al2O3 and only after exceeding a certain Ce-content, AlCeO3 and Ce2O2S get stable. According to FactSage, AlCeO3 is - besides other phases - thermodynamically stable, even for your highest Ce-content. This is why I wonder if you do not find this phase in #4?

According to FactSage, the propsed mechanism in Fig. 10 is therefore not really likely. What is the required Ce-content to reduce Al2O3 according to reaction (8)?